# Role of Carbon Phase in the Formation of Foam Glass Porous Structure

**DOI:** 10.3390/ma15227913

**Published:** 2022-11-09

**Authors:** Boris M. Goltsman, Elena A. Yatsenko

**Affiliations:** Department “General Chemistry and Technology Silicates”, Faculty of Technology, Platov South-Russian State Polytechnic University, 346428 Novocherkassk, Russia

**Keywords:** foam glass, glycerol, thermal treatment, foaming, carbon

## Abstract

The production of durable, non-combustible, heat-insulating materials is currently very important. One of the most promising materials is foam glass. Modern enterprises widely use organic foaming agents in foam glass production. The purpose of this work is to study the role of the carbon phase formed during the organic foaming agent’s (glycerol) thermal destruction in the processes of glass mass foaming. The samples were synthesized using the powder method with high-temperature treatment. Different ratios of glycerol and waterglass in a foaming mixture showed that amount of glycerol should be less than in waterglass. Otherwise, the amount is excessive and the glycerol burns out. It was shown that the quantitative description of the carbon phase structure and properties is complicated by its nanometer size and fusion into the glass. Theoretical calculations demonstrate that carbon particle size cannot be greater than 535 nm. Using a set of methods, it was proved that the carbon phase is represented by nanometer particles of amorphous sp^2^-carbon. Therefore, the foaming mechanism includes nanoparticles settling and immersing into the glass surface, a reaction of carbon with the sulfate ions from glass with a release of gases. Conclusions on foaming intensification via using sulfur additions and other organic foaming agents were drawn.

## 1. Introduction

Foaming of silicate masses is a promising way to obtain effective heat-insulating materials. Such materials are distinguished by durability, resistance to chemical and biological influences, relatively high strength, etc. The porous structure formation could be achieved according to three main mechanisms: the use of foaming additives, so-called “hydrated” or “hydrate” foaming, and self-foaming. Self-foaming technology is used in cases in which the main silicate raw material contains combustible impurities. Such raw materials include, in particular, sewage sludge from municipal wastewater treatment plants, different coal fly ashes, and some others [1,2,3,4]. The technology of “hydrate” foaming is based on the dehydration of alkaline hydro-silicates (either formed by the interaction of NaOH with silicate raw materials or specially introduced into the mixture in the form of waterglass) and foaming of the silicate mass by the released water vapor [5,6,7,8,9,10,11,12,13]. Both of these mechanisms are currently used in laboratory studies and have not been implemented in real production.

Foaming due to the introduction of foaming additives is the most common and most studied method for creating porous silicate materials. The most developed silicate foam material is foam glass produced by large enterprises (FOAMGLAS^®^, Misapor, and others). All industrial technologies and most laboratory research use various types of glass as the main raw material: sheet, bottle, CRT, and others. There are also several studies which use not only glass powder, but also other types of silicate raw materials: sand sludge, red mud, fly ash, enrichment tailings, steel slag, etc. [4,14,15,16,17].

A very wide range of substances is used as foaming additives. These additives are conventionally divided into two types: neutralization and redox. Neutralization foaming agents include, firstly, various carbonates (CaCO_3_, MgCO_3_, Na_2_CO_3_), including such substances as eggshells, sludge from a marble-cutting–polishing plant, reservoir sediments, etc. [12,16,18,19,20,21,22,23,24,25,26,27,28,29]. Redox foaming agents include substances that emit gases as a result of redox reactions. The main substance here is carbon in various modifications: soot, carbon black, carbon ash, activated charcoal, anthracite, graphite, etc. [15,30,31,32,33,34]. It is also possible to use various organic substances (sugar, starch, glycerol, etc.) [12,17,35,36,37,38]. In addition, carbides and nitrides of elements of III and IV groups and even oxides of rare earth metals are used as such foaming agents (Ni_2_O_3_, CeO_2_) [39,40,41,42].

Foam glass enterprises almost always use carbon foaming agents, which is confirmed by the dark color of the resulting material. Moreover, there is a recent trend of replacing carbon black as the foaming agent with organic liquids, especially glycerol. In particular, glycerol is used as a foaming agent by the largest Russian foam glass enterprises: the companies “STES-Vladimir” and “ICM Glass Kaluga” [43,44]. At the same time, studies describing the processes during the glycerol-based foam glass mixtures’ heat treatment are in their initial stages. At present, only the general regularities of the process are described:Interaction of batch components;Release of gases;Formation of pores in a viscous plastic glass mass.

It is important to recognize differences in foam glass synthesis using solid carbon and organic foaming agents. One of the most important previously unstudied points is the function of the carbon phase, which is formed during the thermal decomposition of glycerol and which stains the foam glass a dark color. It is currently not clear if foaming gases are formed directly from glycerol decomposition or if residual carbon also acts as a carbon foaming agent. Such a suggestion is supported by two connected facts:Thermal treatment of organic compounds often leads to their pyrolization and formation of carbon phase [45];Ultrafine carbon (such as carbon black or soot) is an excellent foaming agent in foam glass production [46].

In this regard, the purpose of this work is to study the process of formation of the carbon phase in the foam glass mixture, describe its structure, and analyze its effect on the processes of glass mass foaming.

## 2. Materials and Methods

The main raw material in this study was white (colorless) glass cullet, with a composition of (in wt. %): SiO_2_, 72.0; Al_2_O_3_, 2.4; Fe_2_O_3_, 0.1; CaO, 9.0; MgO, 2.0; Na_2_O, 14.3; K_2_O, 0.1. Water, waterglass (Na_2_O·2SiO_2_·nH_2_O, sodium silicate solution 75Tw), and glycerol (C_3_H_5_(OH)_3_, 98% “AR Grade Fluid”) were used as additives. Preparation of the raw mixture included preliminary grinding of glass to a particle size of less than 5 mm and sequential milling in a laboratory ball mill to a particle size of less than 250 µm (passage through standard sieve No 60); adding glycerol, waterglass, and water (according to developed compositions); and mixing for 10 min to evenly distribute the components [36,38,47,48,49].

The prepared raw mixture was molded into the form of cubes with a face length of 20 mm and a mass of 10 g via uniaxial pressing with a load of 5 MPa. The obtained cubic samples (six samples of each composition) were placed on a mesh stand and loaded into an electric muffle furnace for heat treatment in an atmosphere of air (Figure 1).

Samples were loaded into the furnace at 600 °C. Next, they were heated to the desired foaming temperature of 900 °C at a rate of 10 °C/minute and held at this temperature for 10 min. After foaming, a rapid cooling (quenching) to a temperature of 600 °C at a rate of 20–50 °C/minute was carried out. Then samples were slowly cooled to room temperature (annealed) at a rate of 1–2 °C/minute. After being cooled to 30 °C, samples were removed from the furnace.

Processes of foam glass synthesis and the structure of the resulting samples were recorded using a Canon EOS 550D digital camera (Tokyo, Japan).

Density studies were conducted as follows. Firstly, samples were subjected to mechanical processing to obtain a rectangular form. Then, their linear dimensions were determined with calipers with an accuracy of ±0.1 mm. After that, the volume V of the sample was calculated from its values by multiplying the length of the sample by its width and height. The mass of the samples was measured with an accuracy of 0.01 g. The density of the samples *ρ*, kg/m^3^, was calculated as the ratio of the mass to the volume of the sample according to Equation (1):*ρ* = *m*/*V*·1000, kg/m^3^(1)
where *m* is sample mass, g, and *V* is sample volume, cm^3^.

Each recorded test value is the mean of 5 measurements.

The microstructure of the glass powder and batch was determined using a Quanta200 scanning electron microscope equipped with an EDAX Genesis XVS 30 X-ray microanalysis system (FEI Company, Eindhoven, The Netherlands). The phase composition was determined using a powder X-ray diffractometer ARL (Thermo Fisher Scientific, Walthamm, MA, USA). The equipment is a part of the Collective Use Center “Nanotechnologies” of the Platov South-Russian State Polytechnic University (NPI).

Raman spectroscopy was performed using Renishaw InVia Reflex equipment (Renishaw plc, Wotton-under-Edge, UK). Laser—HeAr (633 nm, 3 mW), lens × 50, N = 0.7. The equipment is a part of the Collective Use Center of the Kurnakov Institute of General and Inorganic Chemistry.

## 3. Results and Discussion

### 3.1. Formation of Carbon Phase

It is known that during the heat treatment of foam glass mixtures, the behavior of glycerol depends on the composition of the foaming mixture. The heating of mixtures from room temperature has been studied previously. It was shown that if the mixture does not contain waterglass, then glycerol evaporates at 190 °C [50]. When the mixture is heated to the foaming temperature, gases are not released, and foaming does not occur. When waterglass is added, the volatilization of glycerol from the sample does not occur, and the process of its decomposition begins at a temperature above 390 °C [50]. This process is extended over time and is externally displayed as the samples’ color changes from white to brown and then to dark gray [51]. This change is due to the incomplete decomposition of glycerol and the formation of what is called “pyrolytic carbon”.

However, in industrial foam glass production, the batch is heated not from room temperature, but from 600–650 °C by loading molded materials into the already-heated furnace. In this regard, the processes occurring during the heat treatment of batches using glycerol foaming agents with loading at 600 °C were studied. For the study, six compositions were prepared (Table 1). Photos of the beginning of heat treatment (30 s after loading) and the coloring of the samples after that are shown in Figure 2.

Figure 2 shows that the absence of waterglass in a batch leads to intensive decomposition of glycerol. For Composition 2 (95 wt. % of glass, 5 wt. % of glycerol) and Composition 3 (91 wt. % of glass, 9 wt. % of glycerol), this is displayed as ignition of a sample and its burning for 40–60 s. In Composition 1 (99 wt. % of glass, 1 wt. % of glycerol), the main part of glycerol evaporates without ignition. Composition 4 (90 wt. % of glass, 1 wt. % of glycerol, 9 wt. % of waterglass) darkens intensely, indicating pyrolytic decomposition of glycerol and formation of carbon. Composition 5 (90 wt. % of glass, 5 wt. % of glycerol, 5 wt. % of waterglass) also turns black; however, due to the larger amount of glycerol, only part of it ignites. The behavior of Composition 6 with an excess of glycerol (90 wt. % of glass, 9 wt. % of glycerol, 1 wt. % of waterglass) is practically similar to Composition 3 (91 wt. % of glass, 9 wt. % of glycerol) and is demonstrated as intense combustion. The decomposition of glycerol in all compositions looks like a narrowing circle associated with gradual heating. Upon complete decomposition, the color of the samples changes from light gray (Composition 1) to almost black (Composition 4). Thus, residual carbon formation occurs in areas with an oxygen shortage. This could be caused by the absence both of air between glass particles and of air inside waterglass, where glycerol is immersed. This indicates the key role of waterglass in the pyrolization of glycerol and the formation of the carbon phase. Further heating to the foaming temperature leads to the following changes in the structure of the samples (Figure 3).

Compositions 1–3 do not contain waterglass, so most of the glycerol is removed (evaporated or/and burned out), while only a small part of it forms the carbon phase. The samples themselves are white, dense, sintered materials consisting of a small number of pores with a size of less than 100 µm (Figure 3). In particular, Composition 1 (99 wt. % of glass, 1 wt. % of glycerol) is a highly melted material with an almost hemispherical shape. Composition 2 (95 wt. % of glass, 5 wt. % of glycerol) has a more rectangular shape, which could be explained by the fact that glycerol vapor slightly foams a melting and settling sample and retains its form. Composition 3 (91 wt. % of glass, 9 wt. % of glycerol) generates even more glycerol vapor, so the shape of the heat-treated sample is almost identical to the initial molded sample. Compositions 4 and 5, where the amount of waterglass is not less than the amount of glycerol, intensively foamed upon reaching the required viscosity. Moreover, Compositions 2 and 3, which initially contained a significant amount of the carbon phase (Figure 2), foamed much more weakly than Composition 4, which contains a smaller amount of glycerol. Composition 6, with the predominance of glycerol (90 wt. % of glass, 9 wt. % of glycerol, 1 wt. % of waterglass) in the foaming mixture, has a low porosity and light gray color with black inclusions [52]. In addition, the external layer color of the Composition 5 sample is lighter than the internal layer color. This is probably caused by carbon burnout due to the reaction of the external part of the sample with oxygen from the atmosphere at a high temperature.

From the obtained results, it can be seen that waterglass inhibits the process of glycerol volatilization and, thus, provides the formation of residual carbon due to glycerol pyrolization, which ensures active foaming of glass mass. However, it is necessary to determine whether this occurs due to the retention of gases from the glycerol decomposition by the waterglass melt or due to the formation of a carbon phase, which plays the role of the main foaming agent.

### 3.2. Influence of Carbon Phase on Glass Foaming

To determine the role of the carbon phase in glass foaming, three compositions were developed. Composition 4 was chosen as the main composition due to having the best foaming activity. Treatment of this composition included mixing the batch, molding the sample, and continually heat treating it from 600 to 900 °C. Composition 4F was obtained based on Composition 4, which was subjected to heat treatment at 600 °C to form the carbon phase. Then, the resulting sample was ground to a particle size of lower than 250 μm to remove gases from the glycerol combustion encapsulated inside the sample. An amount of 3 wt. % of water was added to the resulting powder. Then, samples were molded and heat-treated according to Figure 1. Composition 4F2 was obtained based on Composition 4, with the replacement of 3 wt. % waterglass with the corresponding amount of water (Table 2). This composition can explain the possible effect of added water on foaming. 

The dark color of the obtained Composition 4 samples suggests that they still contain a significant amount of the reactive carbon phase. To check this hypothesis, samples of all three compositions were subjected to secondary foaming. The results of the density measurements are presented in Table 3, those of the internal structure after primary and secondary foaming in Figure 4, and those of the medium pore size distribution in Figure 5. Secondary foaming was carried out both directly on the original samples and on milled and remolded samples. Such milling destroyed the porous structure from primary foaming so we could estimate the secondary foaming ability of the carbon phase.

Figure 4 shows that the structure of all compositions after foaming is very similar regardless of amount of water and synthesis method. After primary foaming, the structure of the samples consists of dark, closed spherical pores of similar size. The density of all compositions also was in the range of 160 kg/m^3^ (Table 3). This indicates the decisive role of the carbon phase in glass mass foaming. After secondary foaming, the color of the samples (regardless of whether they were milled) turns gray-yellow, indicating that a large part of the carbon has burned off. At the same time, original samples subjected to secondary foaming demonstrate an increase in density of 60–80 kg/m^3^, which indicates foam settling due to pore coalescence. This leads to a formation of interconnected pores, so the porosity type shifts from closed to mixed. The same is true for medium pore size distribution (Figure 5). After both primary and secondary foaming of the original samples, a medium pore size was in the range of 0.2–0.8 mm. However, secondary foaming causes 3% of pores to be larger than 1.5 mm, which is observed in Figure 4.

The results of the secondary foaming of samples milled after primary foaming indicate the formation of closed pores by residual carbon with a medium size of 0.1–0.4 mm (Figure 5), and samples with densities in the range of 660–720 kg/m^3^ were obtained. Consequently, the vast majority of the carbon phase reacts during primary foaming of the mixture, and repeated foaming does not lead to significant additional foaming. This may be related to the structure of the carbon phase, which is studied below.

### 3.3. Structure and Properties of Carbon Phase

It is well known that carbon is capable of forming a large number of allotropic modifications both of natural and artificial origins. Their analysis shows that only two types of structure could be formed: graphite and “amorphous” carbon. Graphite has a layered structure; the atoms are bonded more strongly within a layer than between one layer and another. The crystal structure of “amorphous” carbon is identical to the structure of single-crystal graphite. It can be concluded that “amorphous” carbon consists mainly of very small and randomly arranged graphite crystals. The physical properties of “amorphous” carbon depend very much on the dispersion of particles and the presence of impurities. The density, heat capacity, thermal conductivity, and electrical conductivity of “amorphous” carbon are always higher than those of graphite. In nature, amorphous carbon is represented by coke, brown and black coal, soot, etc. The chemical activity of “amorphous” carbon is higher than graphite. Interaction with air oxygen (combustion) occurs at temperatures above 300–500 °C and 600–700 °C with the formation of CO_2_ and CO, respectively.

The presence of the carbon phase in foam glass samples is obvious. The results obtained (color change during heat treatment) clearly indicate this. However, in the study of the carbon phase deposited on glass, several problems can be identified, the main of which are:-The small size of carbon particles (according to calculations below, their size is no more than 535 nm);-A presumably amorphous structure of carbon particles due to their size;-Fusion (immersion) of carbon particles into plastic glass mass during heat treatment.

The thickness of a liquid foaming film and, respectively, carbon phase thickness could be approximately calculated based on batch components’ properties (density, mass, glass particle size). This calculation includes the following assumptions:-Glass particles have a spherical shape;-All particles are of the same size;-Liquid foaming film is evenly distributed on the entire surface.

Glass powder in this study had a particle size lower than 250 μm. Therefore, we can calculate values of the surface, volume, and mass of one spherical glass particle of this diameter using standard geometric and physical formulae (Equations (2)–(4)):*S*_1_*= π·d*^2^, m^2^(2)
*V*_1_*= π·d*^3^/6, m^3^(3)
*m*_1_*=**V*_1_·*ρ*_*g*_, kg(4)
where *d* is particle diameter, m; *S*_1_ is surface of a single glass particle, m^2^; *V*_1_ is the volume of a single glass particle, m^3^; *m*_1_ is the mass of a single glass particle, g; and *ρ_g_* is glass density, 2200 kg/m^3^.

One particle with 250 μm diameter has a surface of 1.96 × 10^−7^ m^2^, a volume of 8.18 × 10^−12^ m^3^, and a mass of 1.8 × 10^−8^ kg. Calculation of liquid film distribution is based on such parameters as specific surface area—a ratio of the dispersed body’s total surface to its mass (cm^2^/g). Therefore, the next step is to calculate the number of particles in 1 g (Equation (5)). Multiplying this value by a single particle’s surface area (Equation (6)), a specific surface area can be calculated:*N = 1/m*_1_;(5)
*S_S_ = N·S*_1_,(6)
where *N* is the number of particles in 1 g; *S_S_* is specific surface area, cm^2^/g.

So, 1 g of glass consists of 55,559 particles with a specific surface area of 109 cm^2^/g. This allows the calculation of a liquid foaming film’s thickness using Equation (7):(7)$$h=\frac{\frac{m_{gl}}{\rho_{gl}}+\frac{m_{wg}}{\rho_{wg}}+\frac{m_{aq}}{\rho_{aq}}}{m_{g}\cdot S_{S}}\cdot 10000,\cdot \mu m$$
where *m_gl_, m_wg_, m_aq_, m_g_* are mass of glycerol, waterglass, water, and glass, respectively, g; *ρ_gl_, ρ_wg_, ρ_aq_* are density of glycerol, waterglass, and water, respectively, g/cm^3^; and *S_S_*—specific surface area, cm^2^/g.

Taking Composition 4F2 (90 wt. % of glass, 1 wt. % of glycerol, 6 wt. % of waterglass, 3 wt. % of water) as a model, masses of glycerol, waterglass, water, and glass for a 10 g cubic sample will be *m_gl_* = 0.1 g; *m_wg_* = 0.6 g; *m_aq_* = 0.3 g; and *m_g_* = 9.0 g, respectively. Substituting these data into Equation (7) foaming film thickness becomes equal to 13.4 μm. It should be emphasized that a real film thickness is many times less due to a higher surface area. This difference is caused by the particles’ irregular (non-spherical) shape and by the fact that 250 μm is the maximum particle size, and D50 for such batches is 118 μm (corresponding *h* is 6.4 μm) [52]. 

Therefore, with the complete transition of glycerol to carbon (which cannot happen) with a glycerol amount in the foaming mixture of 10 wt. % and a carbon amount in glycerol of 40 mol. %, the thickness of the carbon layer will be 535 nm. In practice, of course, the carbon layer thickness is much less, since part of it decomposes to form CO, CO_2_, and other gases.

In this regard, a range of studies was carried out to study the carbon phase. Firstly, electron microscopy of Composition 4 samples after primary and secondary foaming was carried out. The results are shown in Figure 6.

Figure 6 shows the main changes in the internal structure of foam glass at various stages of foaming. Comparing the interpore walls, it can be seen that after primary foaming, the pore walls are smooth, and they contain rounded micropores 4–15 µm in size. After secondary foaming, the pore walls become “rough” and uneven. The pores inside the walls become irregular in shape and decrease to 3–9 µm. The inner surfaces of the pores after secondary foaming also lose the sphericity obtained after primary foaming. This deformation can be explained in two ways. Firstly, softening of the sample during secondary foaming occurs in absence of a foaming agent, and the release of foaming gases forms spherical bubbles–pores. Secondly, the carbon acts as a surface-active agent that stabilizes the glass foam. This means that the decomposition of the carbon phase formed during primary foaming occurs during secondary foaming, which lowers the stability of glass foam [53]. At the same time, it was not possible to detect the carbon phase particles using microscopy due to both their small size and the peculiarities of electron microscopy, which does not allow one to obtain color images. The micrographs show only particles of 1–7 µm in size. These particles are dust from the samples’ mechanical processing during their preparation for the study, partially remaining even after cleaning and washing the prepared samples.

An X-ray phase analysis of the initial glass powder and foam glass after primary and secondary foaming was conducted. The results are presented in Figure 7.

Figure 7 shows that the X-ray diffraction patterns of the samples are very close. A halo in the range of 20–35° indicates a high amount of X-ray amorphous glass phase. Strengthening of the low-intensity peaks at 22 and 30°, corresponding to SiO_2_, and a decrease in the halo height for samples after primary and secondary foaming indicate weak crystallization of the samples associated with repeated heat treatments of the glass powder. Despite partial crystallization, carbon (the amount of which cannot exceed 0.5% (Table 1 and Table 2)) is lost in the “noise” of the amorphous phase of the glass matrix and cannot be reliably identified using this method.

Raman spectroscopy of the initial glass and foam glass after primary heat treatment was carried out. The overview spectrum in the range of 100–4000 cm^−1^ (Figure 8) revealed only two peaks, at 1370 cm^−1^ and 1400 cm^−1^, in the sample of initial glass. The foam glass sample (Composition 4) additionally had a broad peak at 1600 cm^−1^. Therefore, the region around 1400 cm^−1^ was analyzed with a long accumulation time.

Peaks at 1370 cm^−1^ and 1400 cm^−1^ are identical in ratio and shape for both samples and refer to the glass itself. The appearance of a peak at 1600 cm^−1^ in the second sample corresponds to the G-mode of carbon (marked with an asterisk in the figure) [54].

Carbon atoms in the sp^2^-phase can be organized into a highly ordered structure resembling a “honeycomb”. Such a structure, called a graphene layer, can be quite large on an atomic scale. Many such graphene layers form a structure of highly ordered sp^2^ carbon or highly oriented pyrolytic graphite (HOPG). In the Raman spectrum of a graphene monolayer and crystalline graphite, only one vibrational mode at 1581 cm^−1^, called the G-mode, is possible [55]. The G-mode does not necessarily require a hexagonal arrangement of carbon atoms, and its eigenvector corresponds to the movement of pairs of sp^2^ carbon atoms in the plane of the graphene sheet, which may be accompanied by the deformation of carbon “honeycombs”. The G-mode appears whenever there are carbon atoms in the sp^2^-phase.

D-peak in the region of ~1360 cm^−1^ is observed with the appearance of defects in the crystal sp^2^-structure. This mode is forbidden in ideal crystalline graphite and becomes active only in disordered carbon structures in the presence of defects and boundaries of graphene layers. As the number of defects in crystalline graphite increases, the number of carbon atoms located at the boundary of lattice defects and graphene shields increases and the intensity and width of the D-peak also increase significantly. However, if the density of defects reaches the limit, when almost every unit cell in the graphene layer has a defect, a transition to the amorphous sp^2^-phase occurs. In this case, the G- and D-peaks of the Raman spectrum expand so much that they become almost indistinguishable and characterize the sp^2^-carbon in the amorphous phase [56]. Thus, the location, width, and relative intensity of the G- and D-peaks of the Raman spectrum make it possible to quite definitely characterize the phase state of sp^2^-carbon and its structure.

Hence, it can be stated with great confidence that during the thermal treatment of batches with a glycerol foaming agent, the carbon phase is formed due to incomplete decomposition. This phase is represented by amorphous sp^2^-carbon. The most probable mechanism of its formation is the following. High-temperature treatment contributes to the successive breaking of C-H and C-O bonds (starting from the “weakest” bonds with lower energy) with the formation of corresponding gaseous oxides. Instead of broken bonds, chaotic “cross-link” bonds are formed, which then relax from a fragmentary carbon structure into an amorphous solid structure, which consists only of carbon atoms hybridized in the sp^2^-state [57]. The formed amorphous carbon settled on the surface of glass particles and entered into the foaming reactions described below.

The results obtained, despite the impossibility of a quantitative description of the carbon phase, nevertheless qualitatively confirmed its presence and its main properties: an amorphous structure and a nanometer size. 

### 3.4. The Mechanism of the Carbon Phase Action on the Foaming of the Foam Glass Batch

Based on the data obtained, the mechanism of influence of the carbon nanophase on the foaming of glass powder was formulated. The formed carbon amorphous nanoparticles settle on the surface of the glass powder particles. Further heating of the sample leads to a decrease in the viscosity of the glass mass. In this case, the carbon is immersed (fused) into the viscous glass. In addition, the mobility of ions in the glass matrix increases with heating, and the activation energy of various reactions decreases. This allows the carbon to react with the sulfate ions of the glass according to the reaction (8):glass-SO_3_^2−^ + 2C → glass-S^2−^ + CO↑ + CO_2_↑(8)

This leads to the release of gases, which ensures intensive foaming of the softened glass mass. This hypothesis is confirmed by the dynamics of color and structure changes in foam glass samples with an artificially created temperature gradient (Figure 9).

It can be seen from Figure 9 that the color change occurs quite sharply even within the scale of one sample. In particular, the bottom-left corners of both samples are sintered and grey, while the upper-right corners (at a distance of only 20 mm) are melted and beige. This can be explained by the fact that upon reaching a certain glass viscosity, carbon begins to react intensively with it, being oxidized according to Reaction (2) to form CO and CO_2_. In these areas, an increase in volume (foaming) is observed due to the released gases. In addition, there is a color change from dark gray to beige and then (at temperatures above 1000 °C) to orange. As mentioned above, this may be due to the decomposition of carbon at higher temperatures, but this assumption requires further research.

In addition, some effect on foaming is created by: (a)The air enclosed between the glass powder particles;(b)The gases formed during the thermal decomposition of glycerol.

However, as shown above, the main contribution is made by the interaction of the carbon phase and glass mass. An indirect confirmation of the correctness of the described processes is the smell of H_2_S, which is clearly felt organoleptically when pore integrity is violated (sawing, grinding, etc.) [58].

The unreacted carbon remains fused into the glass and gives it a dark color. However, secondary heat treatment leads to very little foaming. The original material after primary foaming shows an increase in density, i.e., foam settling. The absence of expansion during secondary foaming can be explained by the fact that not only carbon is required for gas formation but also sulfate ions from the glass. Since glass based on soda batch was used in this work, it consisted of sulfate ions only in the form of impurities in an amount of less than 0.1%. Therefore, the number of SO_3_ ions in this case should be the limiting factor that determines the completeness of the carbon phase decomposition. It can also be assumed that the use of glass based on a sulfate batch or the additional introduction of sulfur compounds [44] will intensify the foaming.

Another important practical conclusion of this study is the subordinate role of the quality of the initial glycerol raw materials since the main contribution to foaming is made by the carbon nanophase formed during the heat treatment of the organic foaming agent. Therefore, weakly purified technical glycerin or glycerin byproducts of biodiesel production can be used as glycerol foaming agents [59]. Moreover, the data obtained indicate the expediency of continuing the research [36] on the applicability of various organic compounds as foaming agents. This will significantly reduce the cost of foam glass and ensure the recycling of various organic wastes and byproducts.

## 4. Conclusions

The production of foam glass materials using organic foaming agents is a modern technological trend. The purpose of this work was to study the effect of the carbon phase formed during foam glass batches’ thermal treatment with a glycerol foaming agent on the process of glass foaming. If the amount of glycerol exceeds that of waterglass, then glycerol ignites and intensively burns out. Residual carbon is formed in the areas with oxygen shortage (between glass particles and inside waterglass). The decisive role of residual carbon in processes of primary and secondary foaming was discovered. It was shown that the release of foaming gases is almost completely caused by the carbon phase. Primary foaming causes the formation of uniform closed pores with a diameter of 0.5–0.8 mm. Secondary foaming of primary foamed samples leads to a color change from dark gray to gray-yellow, but the increase in volume is very small.

When studying the structure and properties of the carbon phase, several problems associated with its structure and size were revealed. Theoretical calculations proved that the thickness of the carbon phase cannot be more than 535 nm. Based on a complex of studies (X-ray phase analysis, electron microscopy, Raman spectroscopy), the presence of carbon in the form of nanometer particles of amorphous sp^2^-carbon was discovered. The data obtained made it possible to clarify the mechanism of pore formation in foam glass mixtures using glycerol in particular: glycerol decomposition with the formation of a carbon layer on glass particles, immersion of carbon particles in the softened glass, their interaction with sulfate ions from glass with the release of foaming gases, as well as a stabilizing surface-active action of unreacted carbon particles on a viscous glass mass. Practical conclusions about the possibility of foaming intensification by introducing sulfate additives were drawn, as were those about the prospects for using cheaper organic analogs as foaming substances.

## Figures and Tables

**Figure 1 materials-15-07913-f001:**
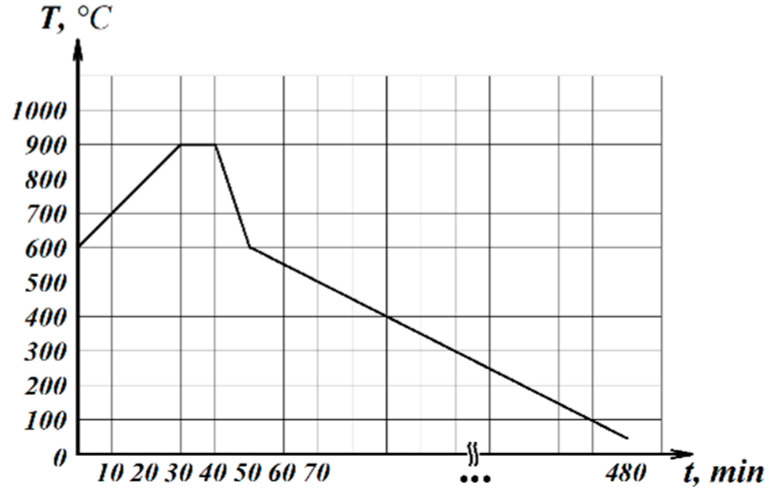
Mode of foam glass batch thermal treatment.

**Figure 2 materials-15-07913-f002:**
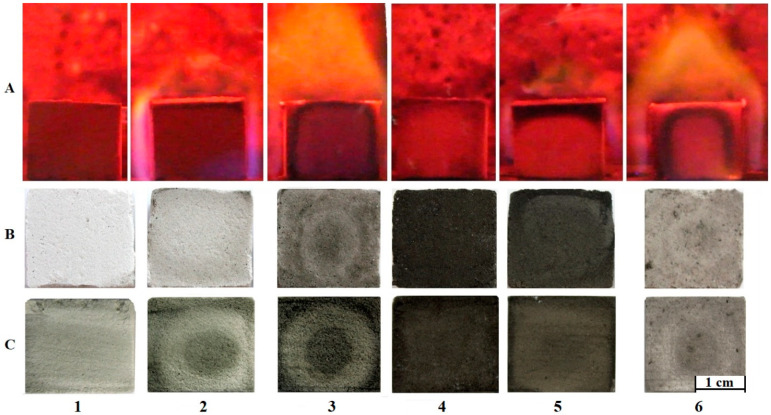
Photos of the first stage of foam glass batches heat treatment: (**A**) dynamics; (**B**) external structure; (**C**) internal structure (cross section); (1) Composition 1 (99 wt. % of glass, 1 wt. % of glycerol); (2) Composition 2 (95 wt. % of glass, 5 wt. % of glycerol); (3) Composition 3 (91 wt. % of glass, 9 wt. % of glycerol); (4) Composition 4 (90 wt. % of glass, 1 wt. % of glycerol, 9 wt. % of waterglass); (5) Composition 5 (90 wt. % of glass, 5 wt. % of glycerol, 5 wt. % of waterglass); (6) Composition 6 (90 wt. % of glass, 9 wt. % of glycerol, 1 wt. % of waterglass).

**Figure 3 materials-15-07913-f003:**
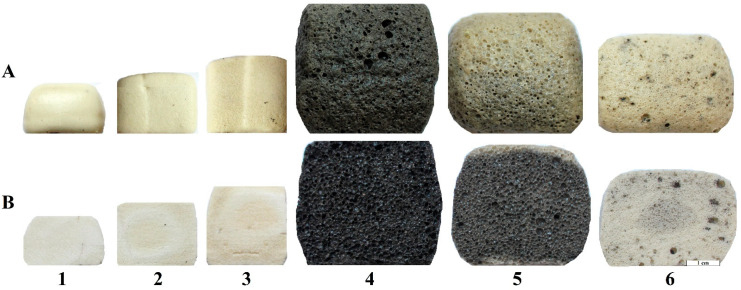
Structure of foam glass samples after foaming: (**A**) external structure; (**B**) internal structure (cross section); (1) Composition 1 (99 wt. % of glass, 1 wt. % of glycerol); (2) Composition 2 (95 wt. % of glass, 5 wt. % of glycerol); (3) Composition 3 (91 wt. % of glass, 9 wt. % of glycerol); (4) Composition 4 (90 wt. % of glass, 1 wt. % of glycerol, 9 wt. % of waterglass); (5) Composition 5 (90 wt. % of glass, 5 wt. % of glycerol, 5 wt. % of waterglass); (6) Composition 6 (90 wt. % of glass, 9 wt. % of glycerol, 1 wt. % of waterglass).

**Figure 4 materials-15-07913-f004:**
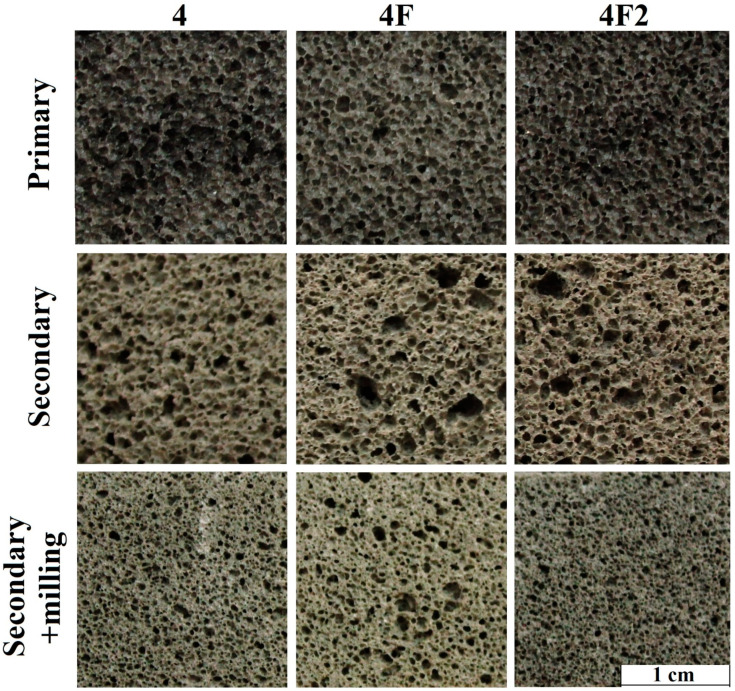
The internal structure of the samples after primary and secondary foaming: (4) Composition 4 (90 wt. % of glass, 1 wt. % of glycerol, 9 wt. % of waterglass); (4F) Composition 4F (90 wt. % of glass, 1 wt. % of glycerol, 9 wt. % of waterglass, and 3 wt. % of water after heat treatment at 600 °C); (4F2) Composition 4F2 (90 wt. % of glass, 1 wt. % of glycerol, 6 wt. % of waterglass, 3 wt. % of water).

**Figure 5 materials-15-07913-f005:**
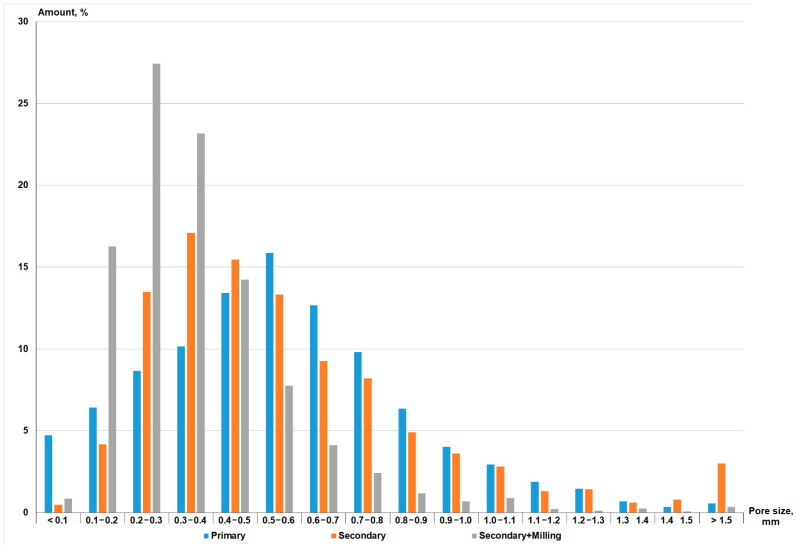
Medium pore size distribution of foam glass samples after primary and secondary foaming.

**Figure 6 materials-15-07913-f006:**
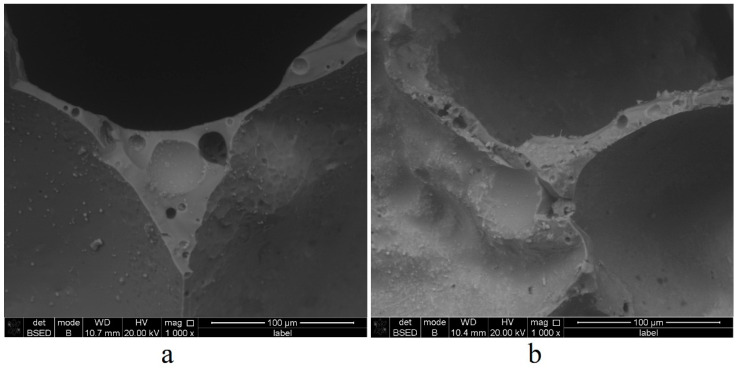
Results of electron microscopy of foam glass samples: (**a**) primary foaming; (**b**) secondary foaming.

**Figure 7 materials-15-07913-f007:**
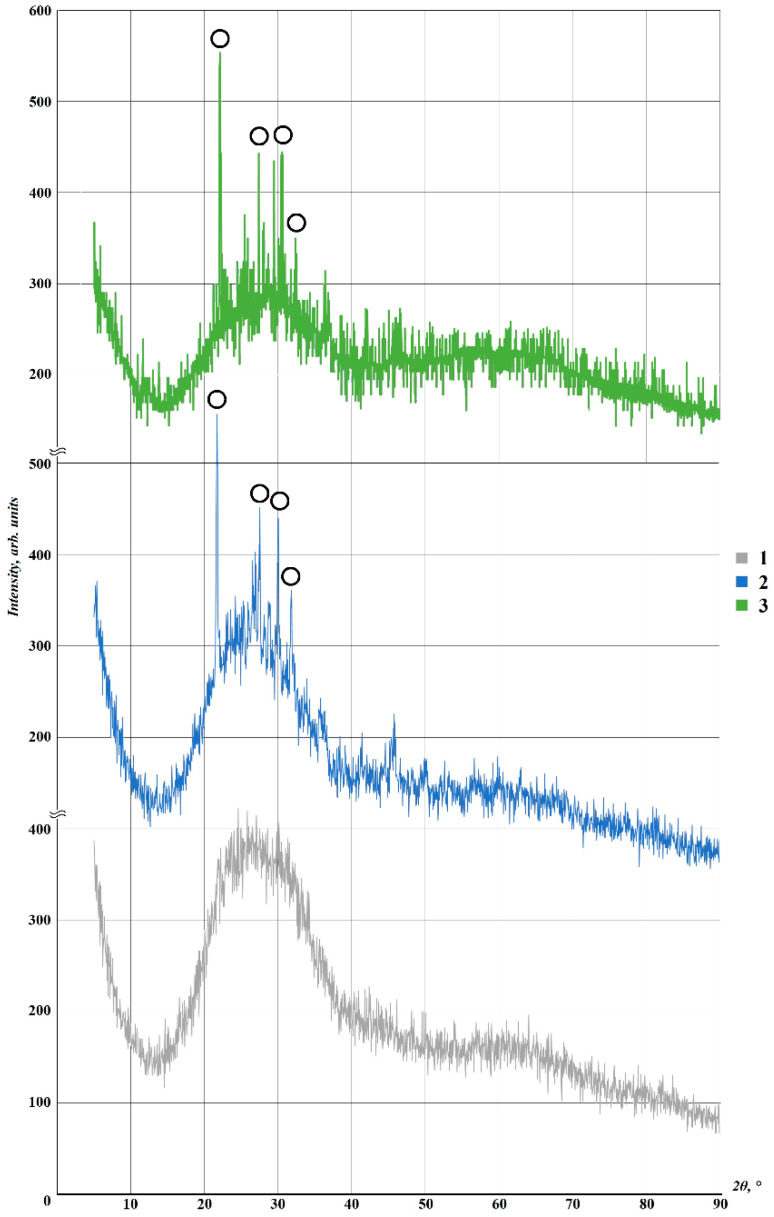
Results of X-ray phase analysis: (1) initial glass; (2) foam glass of Composition 4F2 (90 wt. % of glass, 1 wt. % of glycerol, 9 wt. % of waterglass) after primary foaming; (3) foam glass of Composition 4 after secondary foaming; ○—SiO_2_.

**Figure 8 materials-15-07913-f008:**
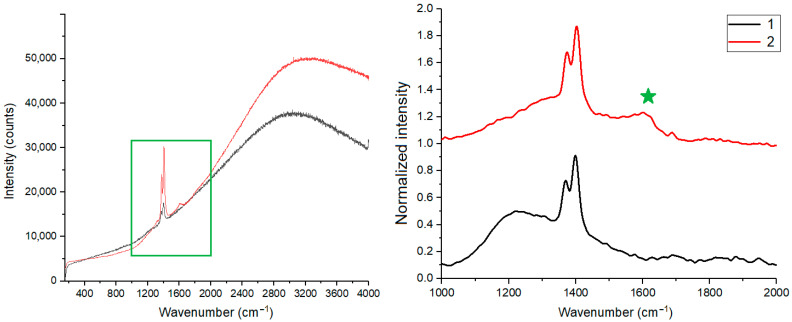
Results of Raman spectroscopy: (1) initial glass, (2) foam glass of Composition 4 (90 wt. % of glass, 1 wt. % of glycerol, 9 wt. % of waterglass) after primary foaming, green box—region of particular analysis, green asterisk—G-mode of carbon.

**Figure 9 materials-15-07913-f009:**
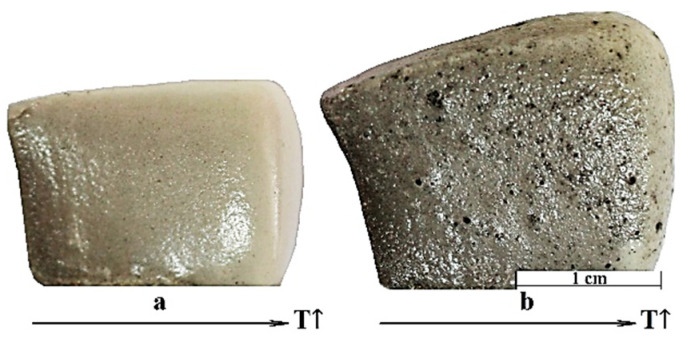
Changes in the structure of foam glass samples under a temperature gradient: (**a**) Composition 3 (91 wt. % of glass, 9 wt. % of glycerol); (**b**) Composition 6 (90 wt. % of glass, 9 wt. % of glycerol, 1 wt. % of waterglass).

**Table 1 materials-15-07913-t001:** Compositions of foam glass batches.

Composition	Amount of the Component, wt. %		
	Glass Powder	Glycerol	Waterglass
1	99	1	-
2	95	5	-
3	91	9	-
4	90	1	9
5	90	5	5
6	90	9	1

**Table 2 materials-15-07913-t002:** Compositions of foam glass batches for secondary foaming.

Composition	Amount of the Component, wt. %			
	Glass Powder	Glycerol	Waterglass	Water
4	90	1	9	-
4F	90	1	9	3 *
4F2	90	1	6	3

* Water was added after heat treatment at 600 °C, as described above.

**Table 3 materials-15-07913-t003:** The density of the samples before and after secondary foaming.

Secondary Foaming	Density, kg/m^3^, of Composition, #		
	4	4F	4F2
Original samples			
Before	161 ± 3	157 ± 7	156 ± 6
After	229 ± 21	238 ± 18	236 ± 20
Milled samples			
Before	860 ± 13	868 ± 14	869 ± 14
After	723 ± 6	659 ± 5	695 ± 7

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
