# Peer review of "Role of Carbon Phase in the Formation of Foam Glass Porous Structure"

_materials, 2022, doi:10.3390/ma15227913_

Round 1

Reviewer 1 Report

The article is very interesting because discusses the role of the carbon phase formed during the thermal destruction of an organic foaming agent (glycerol) in the processes of foaming of glass mass. It is very originality by foam glass materials, Experimental plan and result generating process is well organized. But some modification is needed as follows.

1. The current abstract is an insufficient summary description of the results. Please add the information about the results in the values obtained in the experiment.

2. In section 1, the relevance of the study is not clearly stated. Very briefly, the authors mention the process of formation of the carbon phase in the foam glass mixture. This may justify the relevance of the work.

 3. In section 3.1, line 140: Figure 3 the title of this figure is not in English. Please check.

 4. The authors should have a table with the range of the pore’s diameters after primary and secondary foaming.

 5. Conclusion are not well discussed, and the discussion using different points permits to focus on the most

 6. English should be improved by a native speaker.

Author Response

We thank the Reviewer for the corrections that highly improved our article. We did our best to answer all questions and adjust deficient points of the study, namely:

 The current abstract is an insufficient summary description of the results. Please add the information about the results in the values obtained in the experiment.

The abstract was rewritten, and main results were added (assuming that the maximum length is 200 words).

  1. In section 1, the relevance of the study is not clearly stated. Very briefly, the authors mention the process of formation of the carbon phase in the foam glass mixture. This may justify the relevance of the work.

Information on the carbon phase was added in Section 1.

  1. In section 3.1, line 140: Figure 3 the title of this figure is not in English. Please check.

The title of Figure 3 (line 140) was translated.

  1. The authors should have a table with the range of the pore’s diameters after primary and secondary foaming.

Figure 5 (medium pore size distribution) and its explanation were added to Section 3.2.

  1. Conclusion are not well discussed, and the discussion using different points permits to focus on the most

The conclusion was rewritten.

  1. English should be improved by a native speaker.

English was improved by a native speaker.

Reviewer 2 Report

Relevant research, check suggestions for improvement below. The paper can be shortened a bit. Also, some references can be excluded. 

1.         Abstract: after »… Raman spectroscopy.« there are 5 sentences which doesn’t say anything about the concrete results of this study. This part should be rewritten.

2.         The accuracy of the caliper (±0.1 mm) is much better than the accuracy of the measurement, which is lower due to the irregular (not of rectangular shape) as seen from Fig. 3. Re-estimate the measurement error accordingly.

3.         Most commonly the density is marked be a Greek letter r.

4.         Sentences “ The equipment is a part of the Collective Use Center "Nanotechnologies" of the Platov South-Russian State Polytechnic University (NPI).” and “ The equipment is a part of Collective Use Center of the Kurnakov Institute of General and Inorganic Chemistry.” should be moved to acknowledgements.

5.         “glycerol evaporates at 190 °C”: glycerol boiling point is very high. How can it then evaporate at 190°C? Is the evaporation observed in air and N2, or only in air?

6.         “When waterglass is added, the evaporation of glycerol does not occur”: how does waterglass prevent glycerol evaporation? What is the mechanism?

7.         “in the real foam glass production, the batch is heated not from room temperature, but from 600–650 °C.”: can you explain this in more details? How is the batch heated from room T to 600°C in industry? Usually they use conveyor belt to move the material through the furnace.

8.         “This indicates the key role of waterglass in pyrolization of glycerol and formation of the carbon phase.” Can you explain this mechanism?

9.         Fig. 3 caption is partly in Cyrillic.

10.     “most of the glycerol is removed (evaporated or burned out)”. Which of the mentioned mechanisms is prevailing?

11.     “foaming is not observed. The samples themselves are white dense sintered materials (Figure 3).”: all the samples are foamed, one more others less.

12.     “This is probably caused by carbon burnout due to higher temperature of the external part of a sample.”: despite the insulating properties of the porous sample, the temperature eventually reaches the same temperature throughout the sample. What is the real mechanism behind this?

13.     “waterglass inhibits the process of glycerol removing and residual carbon decomposition.” Again, what is the mechanism behind this?

14.     “due to the retention of gases from the glycerol decomposition by the waterglass melt”: is this really to be expected? How would this actually work?

15.     Line 164-165: probably a mistake, it should be written 3wt% of water glass??

16.     Table 2: correct Cyrillic writing.

17.     Table 2: what is the error in the measurements of the density? Insert the error (e.g. 162+/-X kg/m3)

18.     Section 3.3. Structure and properties of carbon phase: the discussions on the diamond and graphite forms of carbon is not relevant and should be excluded. Only amorphous carbon is expected to form at such conditions. Also Fig. 5 should be removed (it is not a result of the study).

19.     Fig. 7: the curves can be stacked vertically, so the peaks of all samples can be seen clearly. What is evident is that there are some crystalline phases formed. It is not to be expected to identify a small amount of carbon from XRD, even if it would be crystalline, so correct the explanation below the figure.

20.     Fig. 9: it is not possible to observe a (sharp) color change within any of the two samples from this photo.

21.     “In addition, some effect on foaming is created by:

a) the air enclosed between the glass powder particles and

b) the gases formed during the thermal decomposition of glycerol.”

What is the mechanism of option b)? At which temperatures it appears?

22.   “Absence of expansion during secondary foaming can be explained by the fact that not only carbon is required for gas formation, but also sulfate ions from the glass.” Do you really expect that all sulfate ions are consumed during the primary foaming, which is then reflected in absence of secondary foaming? When material is milled after the primary foaming, the carbon becomes exposed to the atmosphere, while before it was protected in the closed pores. Have you measured closed and open porosity of your samples, this would also be good and useful for discussion.

23.     Conclusions should be rewritten according to the changes made in the manuscript.  

Author Response

We thank the Reviewer for the corrections that highly improved our article. We did our best to answer all questions and adjust deficient points of the study, namely:

  1. Abstract: after »… Raman spectroscopy.« there are 5 sentences which doesn’t say anything about the concrete results of this study. This part should be rewritten.

The abstract was rewritten, and main results were added (assuming that the maximum length is 200 words).

  1. The accuracy of the caliper (±0.1 mm) is much better than the accuracy of the measurement, which is lower due to the irregular (not of rectangular shape) as seen from Fig. 3. Re-estimate the measurement error accordingly.

Density was measured using samples that were mechanically processed to obtain a rectangular form. This part of sample preparation was added in the corresponding section.

  1. Most commonly the density is marked be a Greek letter r.

Density marker was changed from d to ρ.

  1. Sentences “ The equipment is a part of the Collective Use Center "Nanotechnologies" of the Platov South-Russian State Polytechnic University (NPI).” and “ The equipment is a part of Collective Use Center of the Kurnakov Institute of General and Inorganic Chemistry.” should be moved to acknowledgements.

Sentences were moved to acknowledgements.

  1. “glycerol evaporates at 190 °C”: glycerol boiling point is very high. How can it then evaporate at 190°C? Is the evaporation observed in air and N2, or only in air?

Glycerol has three characteristic temperatures: flash point (about 180 °C), boiling point (about 290 °C), and auto-ignition temperature (about 390 °C) (see, for example, https://pubchem.ncbi.nlm.nih.gov/compound/Glycerol#section=Flash-Point). Our past results (described in Ref. 47 (doi:10.1007/s10973-020-10015-3)) demonstrated that glass-glycerol mixtures have an endothermic peak at 180-190 °C (glycerol evaporation at flash point temperature). Meanwhile, glass-glycerol-waterglass mixtures have an exothermic peak at 390-395 °C (glycerol combustion at auto-ignition temperature). The link to this study is provided in the article. However, this was a “first bird”, and waterglass-glycerol interaction will be studied later. For example, we already discovered that glycerol decomposition processes do not require oxygen and occur both in air and inert atmosphere.

Answering questions 6-14, we would like to highlight that the “waterglass-glycerol” interaction mechanism was not supposed to be studied in this research work. It is a very interesting and difficult question that will be studied in the following works.

  1. “When waterglass is added, the evaporation of glycerol does not occur”: how does waterglass prevent glycerol evaporation? What is the mechanism?

The term “evaporation” is not quite correct. We meant that glycerol vapor does not volatilize from the sample (described in answer 8). This term was corrected in the text.

  1. “in the real foam glass production, the batch is heated not from room temperature, but from 600–650 °C.”: can you explain this in more details? How is the batch heated from room T to 600°C in industry? Usually they use conveyor belt to move the material through the furnace.

Indeed, industrial molded materials are loaded in the furnace using a conveyor. But the furnace inlet is already heated up to 600-650 °C. So, when a material is entering the furnace, it begins to heat very quickly. This explanation was added to the text.

  1. “This indicates the key role of waterglass in pyrolization of glycerol and formation of the carbon phase.” Can you explain this mechanism?

It can be seen from Fig. 3 that the addition of waterglass leads to the formation of highly porous dark-gray material. As long as there is only glass, glycerol and waterglass in the batch, the dark color could be created only by carbon from glycerol pyrolization. It is supported by comparing compositions 1 (glass and 1 wt. % of glycerol), and composition 4 (glass, 1 wt. % of glycerol and 9 wt. % of waterglass).

So, our hypothesis is as follows. Waterglass “sticks” glass particles together, so glycerol vapor cannot volatilize between them and does not escape from the sample, as it happens in compositions 1-3. Then waterglass forms a melt, and glycerol vapor remains in the sample, surrounded by the glass phase. With heating, it decomposes to gases (CO, CO2, H2O, others), and residual carbon formed due to the lack of oxygen inside the sample (i.e. pyrolization).

  1. Fig. 3 caption is partly in Cyrillic.

The title of Figure 3 (line 140) was translated.

  1. “most of the glycerol is removed (evaporated or burned out)”. Which of the mentioned mechanisms is prevailing?

It depends on the amount of glycerol in the batch. For Compositions with 5 and 9 wt. % of glycerol burning is prevailing, which could be observed in Fig. 2. According to our data, the process happens as follows: glycerol begins to evaporate from the sample, reaches the upper (more heated) part of the furnace, autoignition happens and the whole “column” of glycerol vapor ignites and burns until the evaporation is ended. Hence, these processes are connected, and glycerol burning occurs after evaporation. Slight corrections were added to the text.

  1. “foaming is not observed. The samples themselves are white dense sintered materials (Figure 3).”: all the samples are foamed, one more others less.

Not exactly. We initially meant “foaming” as an increase in the foamed sample’s volume compared to its initial volume. It should be also noted that molded samples consist of about 40-50 % of the air that is encapsulated between glass particles (it is proved by both calculations (Demidovich, Foam glass, 1976, In Russian) and experimental data (for example, the density of glass is about 2200 kg/m3, and the density of the molded sample is about 1250 kg/m3). Thus, compositions 1-3 could be described as follows. Composition 1 is a highly melted material with a hemispherical shape. Composition 2 has a more rectangular shape, which could be explained by the fact that glycerol vapor slightly foams a melting and settling sample and retains its form. Composition 3 generated even more glycerol vapor, so the shape of the heat-treated sample is almost identical to the initial molded sample. But as long as the volume of all three compositions does not exceed the initial molded sample’s volume, we stated that foaming does not occur.

However, we agree that such a statement should be revised, and the paragraph after Fig. 3 was changed.

  1. “This is probably caused by carbon burnout due to higher temperature of the external part of a sample.”: despite the insulating properties of the porous sample, the temperature eventually reaches the same temperature throughout the sample. What is the real mechanism behind this?

We thank the Reviewer for this question. Carbon burnout on the sample’s surface is the process that sometimes occurs in foam glass samples. We discussed this question with specialists in the field, and now we suppose that the main reasons for such color changes are both higher temperatures (due to the gradient between center and surface) and the presence of oxygen at this high temperature (which contacts with carbon and oxidizes it). This explanation is added to the text. And besides, there is a note that this phenomenon will be studied further.

  1. “waterglass inhibits the process of glycerol removing and residual carbon decomposition.” Again, what is the mechanism behind this?

This sentence was changed to “… waterglass inhibits the process of glycerol volatilization and, thus, provides the formation of residual carbon due to glycerol pyrolization”. Our hypothesis is described in answer # 8.

  1. “due to the retention of gases from the glycerol decomposition by the waterglass melt”: is this really to be expected? How would this actually work?

This hypothesis is described in answer # 8. And again, we want to stress that these thoughts have some experimental background, but they will be studied in the following works.

  1. Line 164-165: probably a mistake, it should be written 3wt% of water glass??

No, this line is correct. We added 3 wt. % of water to make a powder mixture able to mold. And afterward, we prepare 4F2 composition to study any possible effect of water on the mixture’s foaming.

  1. Table 2: correct Cyrillic writing.

Cyrillic writing was corrected.

  1. Table 2: what is the error in the measurements of the density? Insert the error (e.g. 162+/-X kg/m3)

Error bars were added.

  1. Section 3.3. Structure and properties of carbon phase: the discussions on the diamond and graphite forms of carbon is not relevant and should be excluded. Only amorphous carbon is expected to form at such conditions. Also Fig. 5 should be removed (it is not a result of the study).

The first paragraph of section 3.3 was shortened and rewritten. Fig. 5 was also removed.

  1. Fig. 7: the curves can be stacked vertically, so the peaks of all samples can be seen clearly. What is evident is that there are some crystalline phases formed. It is not to be expected to identify a small amount of carbon from XRD, even if it would be crystalline, so correct the explanation below the figure.

We couldn’t know for sure that the carbon phase won’t be represented on XRD curves, so we checked it. The appearance of crystalline phases is explained in the text by partial recrystallization of the glass matrix. A description of peaks was added in Fig. 7.

  1. Fig. 9: it is not possible to observe a (sharp) color change within any of the two samples from this photo.

The figure was processed to improve the visibility of described changes. Also, the description of these changes was added to the text.

  1. “In addition, some effect on foaming is created by:
  2. a) the air enclosed between the glass powder particles and
  3. b) the gases formed during the thermal decomposition of glycerol.”

What is the mechanism of option b)? At which temperatures it appears?

As described in answer #5, glycerol has three characteristic temperatures, and its evaporation depends on the batch composition. In any case, thermal treatment of glycerol above 600 °C leads to its decomposition as follows:

- if there is enough oxygen (sample surface) then it decomposes according to the standard organic equation (organic compound + O2 = CO2 +CO + H2O).

- if there is a lack of oxygen (sample core) then it pyrolyzed according to the formula: CmHnOk → CxHyOz + gas (H2,CO,CO2,CH4,...) + Coke (T. Valliyappan et al. / Bioresource Technology 99 (2008) 4476–4483)

So, option b) suggests the above-mentioned gases encapsulated in the viscous glass.

  1. “Absence of expansion during secondary foaming can be explained by the fact that not only carbon is required for gas formation, but also sulfate ions from the glass.” Do you really expect that all sulfate ions are consumed during the primary foaming, which is then reflected in absence of secondary foaming? When material is milled after the primary foaming, the carbon becomes exposed to the atmosphere, while before it was protected in the closed pores. Have you measured closed and open porosity of your samples, this would also be good and useful for discussion.

The reaction of carbon with sulphate-ions seems the most probable mechanism because of the following reasons. 1. Milling indeed leads to a stronger contact of carbon and air. However, samples, which were secondary foamed but weren’t milled, show the same color change as the milled ones. So, the reason for this change is inside samples. 2. Color change is obviously connected with the carbon phase, in particular, its reaction with an oxidizing compound (which could be in glass and/or in a gas atmosphere in pores). As described in answer # 21, pore gases are H2, CO, and others, i.e. the atmosphere is not oxidizing. Thus, the only explanation is in the reaction of carbon and some component of glass. Considering both previous researches of other authors and our experimental data, the most probable oxidizing component in glass is sulphate-ions from glass. This hypothesis is supported by a) the smell of H2S after breaking pore walls, b) the well-known application of carbon as a reducing agent in glass melting (where it also reacts with Na2SO4) and c) the addition of elemental Sulphur as a batch component at foam glass factories (for example, Russian Federation Patent 2745544) where it acts as an oxidizing agent.

Speaking of porosity type – our research didn’t have this aim. But we analyzed this aspect and added the corresponding Section 3.2.

  1. Conclusions should be rewritten according to the changes made in the manuscript.

Conclusions were rewritten.

Reviewer 3 Report

General remarks 

Revise English in the following sentences: 

  • Lines 56-58: “In particular, glycerol is used as a foaming agent by the largest Russian foam glass enterprises. – “Company “STES-Vladimir” and “ICM Glass Kaluga””  

  • Line 84: “sharp cooling” - check if “sharp” is the correct technical term 

  • Line 116: " in the real foam glass production, " check if “real” is the correct technical term (I recommend using industrial or commercial) 

  • Line 119: “6 compositions were developed” check if “developed” is the correct technical term (I recommend using prepared or produced) 

  • Line 131: “only part of it ignites” check if “ignite” is the correct technical term 

  • Lines 158-159: “three compositions were obtained.” check if “obtained” is the correct technical term 

  • Lines 182-184: “The density of Composition 4 samples was 162 kg/m3, Composition 4F – 157 kg/m3, Composition 4F2 – 156 kg/m3.” 

Each figure must be clear and understandable on its own without searching for the relevant text. However, the figures are incomplete, as the composition of the samples is missing information from the captions. 

Remarks to figures 

Figs 2-3-4-6-8-9: Add an explanation, of what the numbers at the bottom of the image represent. In the caption, add the details of the composition. 

Fig 3: Change the language of the caption to English 

Figure 7: As the graphs cover each other, the lines should be shifted.  

Materials and Methods  

Line 70: correct as follows: glass cullet, with a composition (in wt. %): SiO2 

Line 72: “silicate solution 75Tw)” – correct it 

Line 72: add the formula of the glycerol 

Line 73: “mixture included grinding” -describe the grinding process as well 

Line 74: “adding glycerol, waterglass, and water” - add the amount of water 

Lines 77-78: “the obtained cubic samples” - add, how many samples were prepared  

Line 94: “m –sample” a space is missing 

Add to the methods, how were the photos seen in Figs 1-2 taken?  

Results and Discussion 

Table 1. Change content to amount 

Lines 126-138: “Compositions 2, 4, etc” - besides the code, add information about the composition of the samples. It makes it hard to follow the paper if somebody always must turn pages back and fore during the reading of the text (same for the figures). This remark is valid for the whole text, where only Composition # is mentioned. 

Add the composition of samples 4F and 4F2 to Table 1 as well. 

Add to the methods, how were the photos seen in Fig.4 taken? 

Line 173: “as is” is not a suitable word for the academic style. Instead of this, use initial or original (same for Line 187) 

Table 4. change Cyrill letters to Latin, “as is” to initial or original 

Line 180: “The Figure 4 shows” correct form: Figure 4 shows 

The first paragraph of chapter 3.3 is irrelevant. 

Line 218: “small size of carbon particles (according to previous calculations,” - this calculation is not described in the methods, add it. If it is described elsewhere, refer to that paper.   

Line 228: “The Figure 6 shows” correct form: Figure 6 shows 

Lines 239-241: “At the same time, it was not possible to detect the carbon phase particles using microscopy due to both their small size and peculiarities of electron microscopy, which does not allow obtaining color images” - what about the elemental analysis? 

Line 241: “microphotographs” correct form: micrographs 

Line 250: “The Figure 7 shows” correct form: Figure 7 shows 

Line 260: “in the first sample at 1370 cm-1 and 1400 cm-1. The second sample” define, which one (composition, code, etc) is the first and second sample 

Line 298: “in the sp2-state.” remove the dot 

Lines 305-306: “According to the calculations, the average thickness” this calculation is not described in the methods, add it. If it is described elsewhere, refer to that paper.   

Lines 375-376: “Practical conclusions were made” Detail, what conclusions were made 

References 

Ref 25: “Effect of Na2CO3 as Foaming Agent” add subscription 

Ref 46: “Rome, Italy” letter size is 12 instead of 11 

Author Response

We thank the Reviewer for the corrections that highly improved our article. We did our best to answer all questions and adjust deficient points of the study, namely:

General remarks 

Revise English in the following sentences: 

  • Lines 56-58: “In particular, glycerol is used as a foaming agent by the largest Russian foam glass enterprises. – “Company “STES-Vladimir” and “ICM Glass Kaluga”” 

The line was changed.

  • Line 84: “sharp cooling” - check if “sharp” is the correct technical term

The term “sharp” was replaced by “rapid”.

  • Line 116: " in the real foam glass production, " check if “real” is the correct technical term (I recommend using industrial or commercial)

The word “real” was changed to “industrial”.

  • Line 119: “6 compositions were developed” check if “developed” is the correct technical term (I recommend using prepared or produced)

The word „developed” was changed to „prepared”.

  • Line 131: “only part of it ignites” check if “ignite” is the correct technical term

We suppose that “ignite” is a correct term, as long as it occurs above the glycerol autoignition temperature (for example, https://pubchem.ncbi.nlm.nih.gov/compound/Glycerol#section=Autoignition-Temperature).

  • Lines 158-159: “three compositions were obtained.” check if “obtained” is the correct technical term

The term “obtained” was replaced by a more suitable term in this line „developed”.

  • Lines 182-184: “The density of Composition 4 samples was 162 kg/m3, Composition 4F – 157 kg/m3, Composition 4F2 – 156 kg/m3.”

The sentence was rewritten.

Remarks to figures 

Figs 2-3-4-6-8-9: Add an explanation, of what the numbers at the bottom of the image represent. In the caption, add the details of the composition. 

Details on the numbers and compositions were added.

Fig 3: Change the language of the caption to English 

The title of Figure 3 (line 140) was translated.

Figure 7: As the graphs cover each other, the lines should be shifted.  

Figure 7 was changed (lines were shifted).

Materials and Methods  

Line 70: correct as follows: “glass cullet, with a composition (in wt. %): SiO2 

The line was corrected.

Line 72: “silicate solution 75Tw)” – correct it 

The line was corrected.

Line 72: add the formula of the glycerol 

The formula was added.

Line 73: “mixture included grinding” -describe the grinding process as well 

Grinding-milling process description was extended.

Line 74: “adding glycerol, waterglass, and water” - add the amount of water 

We suppose that adding a specific amount of water (as well as glycerol and waterglass) isn’t quite suitable because it depends on the specific composition. This note was added to the text.

Lines 77-78: “the obtained cubic samples” - add, how many samples were prepared  

The information was added.

Line 94: “m –sample” a space is missing 

A space was added.

Add to the methods, how were the photos seen in Figs 1-2 taken?  

The information was added.

Results and Discussion 

Table 1. Change content to amount 

The term was changed throughout the text.

Lines 126-138: “Compositions 2, 4, etc” - besides the code, add information about the composition of the samples. It makes it hard to follow the paper if somebody always must turn pages back and fore during the reading of the text (same for the figures). This remark is valid for the whole text, where only Composition # is mentioned. 

Information on the compositions was added throughout the text.

Add the composition of samples 4F and 4F2 to Table 1 as well. 

Compositions 4, 4F, 4F2 were added to a new Table 2 to prevent “page turning” that Reviewer asks to avoid in the remark above.

Add to the methods, how were the photos seen in Fig.4 taken? 

The information was added.

Line 173: “as is” is not a suitable word for the academic style. Instead of this, use initial or original (same for Line 187) 

The phrase “as is” was replaced with the term “original”.

Table 4. change Cyrill letters to Latin, “as is” to initial or original 

The changes were added.

Line 180: “The Figure 4 shows” correct form: Figure 4 shows 

The line was corrected

The first paragraph of chapter 3.3 is irrelevant. 

The first paragraph was shortened and rewritten.

Line 218: “small size of carbon particles (according to previous calculations,” - this calculation is not described in the methods, add it. If it is described elsewhere, refer to that paper.   

The calculation of carbon particles size was added to Section 3.3.

Line 228: “The Figure 6 shows” correct form: Figure 6 shows 

The line was corrected.

Lines 239-241: “At the same time, it was not possible to detect the carbon phase particles using microscopy due to both their small size and peculiarities of electron microscopy, which does not allow obtaining color images” - what about the elemental analysis? 

We tried to implement different types of elemental analysis for our aims, but didn’t succeed. Thus, EDS-analysis was unsuitable for it, because it has a big error bar when measuring elements with atomic weight less than sodium. XRF-analysis cannot differentiate carbon and LOI compounds, and calculating carbon as LOI isn’t right, because losses also include water (including water from waterglass) and small combustible impurities. So, these results couldn’t be represented as carbon amount.

Line 241: “microphotographs” correct form: micrographs 

The term was corrected.

Line 250: “The Figure 7 shows” correct form: Figure 7 shows 

The line was corrected.

Line 260: “in the first sample at 1370 cm-1 and 1400 cm-1. The second sample” define, which one (composition, code, etc) is the first and second sample 

Clarifications on „first” and „second” samples were added.

Line 298: “in the sp2-state.” remove the dot 

The dot was removed.

Lines 305-306: “According to the calculations, the average thickness” this calculation is not described in the methods, add it. If it is described elsewhere, refer to that paper.   

Calculations of carbon particles size were added to Section 3.3.

Lines 375-376: “Practical conclusions were made” Detail, what conclusions were made 

Details were added.

References 

Ref 25: “Effect of Na2CO3 as Foaming Agent” add subscription 

Ref 46: “Rome, Italy” letter size is 12 instead of 11 

References were corrected.